# Antioxidant and Cytotoxic Evaluation of Aqueous Extracts from *Hymenochaetaceae* Fungi Associated with Endemic Chilean Sclerophyll Forest Trees

**DOI:** 10.3390/ijms26125877

**Published:** 2025-06-19

**Authors:** Suleivys M. Nuñez, Ahyra García, Tanya Roman, Luis Aguilar, María Elena Tarnok, Fanny Guzmán, Constanza Cárdenas, Sebastián Ponce, Dreidy Vásquez, Samuel Carrasco, José Luis Valín

**Affiliations:** 1Escuela de Ingeniería Química, Pontificia Universidad Católica de Valparaíso, Valparaíso 2340025, Chile; dreidy.vasquez@pucv.cl (D.V.); samuel.carrasco@pucv.cl (S.C.); 2Instituto de Química, Pontificia Universidad Católica de Valparaíso, Valparaíso 2373223, Chile; ahyra.garcia@pucv.cl (A.G.); luis.aguilar@pucv.cl (L.A.); maria.tarnok@pucv.cl (M.E.T.); 3Programa de Doctorado en Biotecnología, Pontificia Universidad Católica de Valparaíso/Universidad Técnica Federico Santa María, Valparaíso 2340025, Chile; tanya.roman.b@mail.pucv.cl; 4Núcleo de Biotecnología Curauma (NBC), Pontificia Universidad Católica de Valparaíso, Valparaíso 2373223, Chile; fanny.guzman@pucv.cl (F.G.); constanza.cardenas@pucv.cl (C.C.); 5Programa de Magister en Ciencias Biológicas, Mención Biodiversidad y Conservación, Facultad de Ciencias, Universidad de Valparaíso, Gran Bretaña 1111, Valparaíso 2360102, Chile; sebastian.ponce@postgrado.uv.cl; 6Club de Micología y Estudios Agroecológicos de la Provincia de Marga Marga, Manuel Rodríguez 128, Limache 2240229, Chile; 7Escuela de Ingeniería Mecánica, Pontificia Universidad Católica de Valparaíso, Valparaíso 2430120, Chile; jose.valin@pucv.cl

**Keywords:** *Hymenochaetaceae* fungi, aqueous extracts, *Inonotus* sp., *Fulvifomes* sp., *Phylloporia boldo*, sclerophyll forest, antioxidant activity, polyphenols, cytotoxicity, in vitro assays

## Abstract

In the search for safe and effective natural antioxidants, this study investigates the antioxidant and cytotoxic properties of aqueous extracts obtained from three fungi of the family *Hymenochaetaceae*: *Inonotus* sp., *Fulvifomes* sp., and *Phylloporia boldo*, all associated with endemic trees of the Chilean sclerophyll forest. Antioxidant capacity was assessed through DPPH, ABTS, and hydroxyl radical scavenging assays. *Fulvifomes* sp. exhibited the highest antioxidant activity across all methods, which was consistent with its elevated polyphenol content. *P. boldo*, on the other hand, had the highest protein concentration but comparatively lower antioxidant activity. Cytotoxicity was evaluated using the WST-1 assay in the RTgill-W1 salmonid cell line, revealing that *Inonotus* sp. displayed the lowest cytotoxicity at both tested concentrations, suggesting it may be suitable for bioactive applications in aquaculture. In contrast, *Fulvifomes* sp. and *P. boldo* showed significant cytotoxic effects at higher concentrations. These findings highlight the potential of *Inonotus* sp. as a natural antioxidant with low cytotoxicity and encourages further exploration of native forest fungi as sources of functional bioactive compounds for food, nutraceutical, or aquaculture applications.

## 1. Introduction

Reactive oxygen species (ROS) are highly reactive molecules and radicals widely present in living organisms. While they are essential at low concentrations for signaling and defense mechanisms, excessive ROS accumulation causes oxidative stress, damaging cellular components such as proteins, lipids, and DNA [1]. This imbalance is associated with various pathological conditions, including aging, cancer, neurodegenerative disorders, and cardiovascular diseases [2]. Consequently, there is a growing interest in identifying natural antioxidant compounds capable of mitigating ROS-related damage.

Among the natural sources of bioactive molecules, fungi have emerged as promising candidates due to their ability to produce a wide range of secondary metabolites with antioxidant, antimicrobial, and anticancer potential. In particular, fungi from the *Hymenochaetaceae* family, a group of wood-decaying basidiomycetes, are abundant in forest ecosystems and play a key ecological role in organic matter decomposition. Recent research has shown that members of this family possess high concentrations of phenolic compounds, polysaccharides, and terpenoids, contributing to their biological activity [3].

In Chile, the sclerophyll forest ecosystem, located mainly in the central Mediterranean zone, represents a biodiversity hotspot with high levels of endemism. Native trees such as *Quillaja saponaria*, *Schinus latifolius*, and *Peumus boldus* support unique ecological interactions and serve as hosts to various fungal species, including *Inonotus aff. quercustris* (*“aff.”* indicating a species affine to *Inonotus quercustris*), *Fulvifomes* sp. (“sp.” indicates that the specific species has not been determined), and *Phylloporia boldo*. These fungi, associated with endemic hosts, may harbor distinctive antioxidant compounds shaped by their adaptation to oxidative environments associated with wood degradation [4,5]. However, their bioactive potential remains poorly explored.

The antioxidant mechanisms of fungi include chain reaction inhibition, metal ion chelation, and radical scavenging [6]. Studies on Basidiomycota have shown that the concentration of phenols correlates directly with their antioxidant capacity [7,8]. Furthermore, Chilean basidiomycetes have demonstrated the ability to synthesize active compounds against microorganisms, opening new opportunities for antimicrobial agent development [9].

Within the *Hymenochaetaceae* family, several genera of wood-decomposing fungi are distinguished, such as *Inonotus*, *Fulvifomes*, and *Phylloporia*. These fungi share common characteristics but are differentiated by specific morphological and genetic features that may relate to interactions with their hosts [3]. *Inonotus obliquus*, for instance, has shown anticancer, anti-inflammatory, and hypoglycemic activity [10], and its aqueous extracts have demonstrated high antioxidant capacity [11]. For *Phellinus rimosus*, efficacy against reduced antioxidant states and hyperglycemia associated with diabetes has been verified [12].

The isolation of phenolic compounds, such as orgone from *Phellinus fastuosus*, has shown it to be a potent cytotoxic agent against cancer cell lines RD and HepG-2 while less toxic to normal CC-1 cells. However, there has been little research on its antimicrobial or antioxidant properties [13]. Some species within the genus *Phylloporia* often have a particular relationship with their hosts, as seen with *P. boldo* and its host, *Peumus boldus*. Polysaccharides derived from *Peumus boldus* have been attributed with anticancer potential against HCT-116 carcinoma, MCF-7 breast adenocarcinoma, and HL-60 leukemia cell lines [14]. A study by Zhao and collaborators detected 46 antioxidant compounds in *P. ribis* [15]. Macrofungi growing on dead or live tree must degrade it, and this exposure to oxidative stress makes them candidates for antioxidant research and further exploration of their potential through additional studies.

Understanding the antioxidant and cytotoxic properties of these fungi is relevant not only for ecological and taxonomic purposes but also for their potential applications in health, food, and aquaculture industries. The study of aqueous extracts, in particular, aligns with traditional preparation methods and allows for the recovery of polar compounds, which are often responsible for antioxidant activity.

Therefore, this study aims to evaluate the antioxidant activity and cytotoxicity of aqueous extracts from three *Hymenochaetaceae* fungi, *I. aff. quercustris*, *Fulvifomes* sp., and *P. boldo*, collected from endemic trees of the Chilean sclerophyll forest. By integrating chemical, biological, and cellular assays, this research seeks to provide new insights into the bioactive potential of these fungi and to highlight their relevance as natural sources of antioxidants with possible applications in functional foods, nutraceuticals, and aquaculture.

## 2. Results and Discussion

### 2.1. Characterization of Aqueous Extract

#### 2.1.1. Determination of Total Polyphenols and Proteins

The aqueous extraction method, applied for 1 h and supplemented with 10 min of sonication, has proven effective in extracting proteins and polyphenols. Proteins, being large and primarily water-soluble molecules, are efficiently extracted under these conditions. In mushrooms, proteins perform structural, enzymatic, and other essential biological roles, suggesting a higher concentration than polyphenols. On the other hand, polyphenols act as chemical defenses and antioxidants [16]. Sonication also significantly improves polysaccharide recovery, which, although not measured in this study, could contribute to the observed antioxidant activity [17]. These findings emphasize the relevance of the extraction method used and the importance of bioactive compounds in mushrooms, suggesting their potential application in the food and pharmaceutical industries.

Although relevant results were obtained for each of the analyzed extracts, it is essential to note that the taxonomic identifications of two of the species (*Inonotus* sp. and *Fulvifomes* sp.) have not yet been resolved at the species level. This limitation, derived from the taxonomic complexity of the genera involved, does not invalidate the results obtained, but it does affect their future reproducibility. Molecular sequences are included as support, and further studies are encouraged to delve deeper into the identification and taxonomic characterization of these fungi.

The polyphenol content of each extract was quantified using the Folin–Ciocalteu method and expressed in terms of gallic acid equivalents (GAEs). The results are presented in Figure 1a, and show the following polyphenol distribution: *Fulvifomes* sp. > *P. boldo* > *Inonotus* sp. *Fulvifomes* sp. showed the highest polyphenol concentration, reaching 63.28 ± 2.53 mg GAE/g, followed by *P. boldo* with 47.08 ± 1.24 mg GAE/g, while *Inonotus* sp. showed the lowest concentration, with 33.00 ± 2.08 mg GAE/g.

In terms of protein measurements (Figure 1b), *P. boldo* showed the highest concentration with 229.70 ± 1.3 mg/g (22.9%), followed by *Fulvifomes* sp. with 191.33 ± 1.6 mg/g (19.13%), and *Inonotus* sp. had the lowest protein concentration at 151.80 ± 1.9 mg/g (15.15%). According to González et al. [18], edible mushrooms typically contain protein levels ranging from 18.87% to 36.96%, aligning with previously reported ranges for basidiomycetes of 10.53% to 23.29% [19].

Previous studies, such as Prasad and collaborators [20], report that the total polyphenol content (TPC) in fruiting bodies of basidiomycete mushrooms ranges between 6.08 and 24.85 mg GAE/g. Aqueous extraction of polyphenols in three edible mushrooms has also been shown to be more efficient than ethanol extraction, yielding polyphenol concentrations up to three times higher: 7.96 mg GAE/g, 9.93 mg GAE/g, and 10.56 mg GAE/g in *Flammulina velutipes*, *Pleurotus pulmonarius*, and *Pleurotus eryngii*, respectively [21]. Although antioxidant activity can vary among mushrooms from different taxonomic groups and ecological niches, these results highlight the effectiveness of aqueous extraction and the potential of mushrooms to provide significant bioactive compounds [22].

#### 2.1.2. Identification of Phenolic Compounds

The phenolic profile of the aqueous fungal extracts was analyzed by HPLC-FLD using selected standards commonly reported for fungi of the *Hymenochaetaceae* family [23,24], including gallic acid, catechin, epicatechin, syringic acid, caffeic acid, and p-coumaric acid.

The results obtained (Figure 2 and Figure 3) show that, in general, the identification of polyphenols in the analyzed samples was limited. In Figure 2C and Figure 3C for the *Fulvifomes* sp. extract sample, the presence of syringic acid is identified with certainty, while the possible presence of gallic acid is suggested, although without definitive confirmation. In the *P. boldo* extract sample (Figure 2B and Figure 3B), syringic acid was also identified, but other compounds of interest were not detected. Finally, in the *Inonotus* sp. extract sample (Figure 2D and Figure 3D), none of the analyzed polyphenols could be identified, suggesting that either their concentration is below the detection limit of the method used or that the compounds present in the sample do not match the selected standards. A summary of these results is shown in Table 1.

In particular, the identification of syringic acid in the extract samples of *Fulvifomes* sp. and *P. boldo* agrees with the retention times observed in the chromatograms at 360 nm (Figure 2B,C), and 422 nm (Figure 3B,C), the only compound that could be confirmed with certainty. The rest of the compounds analyzed did not present coincidences in their retention times, suggesting that they might not be present in the samples or that the experimental conditions limited their detection. Furthermore, when comparing the results obtained with the standards, it was observed that syringic acid is the least hydrophobic of the compounds detected. This could explain its better solubility and detection in the aqueous extract. These results are consistent with the highest concentration of polyphenols present *in Fulvifomes* sp. and *P. boldo* (Figure 1a).

It should be noted that although only one polyphenol was identified, other phenolic compounds in the samples cannot be ruled out. Some polyphenols present may not be identifiable with the method used due to limitations in sensitivity or lack of agreement with the standards used. In future studies, it would be advisable to explore complementary analytical methods, such as mass spectrometry, for a more complete identification of the phenolic profile of the extracts. 

To verify that the peaks identified in the extracts with the same retention times of syringic acid correspond to this polyphenol, the fluorescence spectra extracted from the corresponding chromatograms were compared; these spectra are shown in Figure 4. It can be observed that the spectra are very similar, with some differences that can be attributed to spectral overlapping with other compounds that elute at the same retention time.

### 2.2. Antioxidant Activity

#### 2.2.1. DPPH Scavenging Assay

The antioxidant activity of the three aqueous extracts was evaluated using the DPPH method at a fixed concentration of 3 mg/mL (Figure 5a). Among them, *Fulvifomes* sp. demonstrated the highest radical scavenging activity (65%), followed by *P. boldo* (36%) and *Inonotus* sp. (24%). For comparison, vitamin C at the same concentration of 3 mg/mL, exhibited a markedly higher activity of 96%. This trend in antioxidant performance is consistent with the polyphenol content measured in each extract.

The aqueous extract from *Fulvifomes* sp., which exhibited the highest antioxidant activity among the tested samples, was selected for a comparative analysis against several positive controls. These included vitamin C, trolox, gallic acid, and two commercial products: a black tea with blueberries and maqui (Tea) and a collagen supplement enriched with vitamin C and magnesium (Plus). All samples were tested at concentrations ranging from 0.1875 to 6 mg/mL using a two-fold serial dilution (Figure 5b). This extract was chosen due to its superior antioxidant performance, with the goal of assessing how different concentrations affect its radical scavenging capacity. As expected, the standard antioxidants (vitamin C, trolox, and gallic acid) demonstrated consistently high DPPH radical scavenging activity.

The *Fulvifomes* sp. extract together with Tea, and the Plus supplement, exhibited significantly lower antioxidant activity than standard controls at lower concentrations. However, *Fulvifomes* sp. extract demonstrated superior results to Tea and Plus. A trend of increasing antioxidant activity with concentration was observed, reaching a maximum value at 1.5 mg/mL with a DPPH scavenging value of 83%, comparable to standard controls, with no statistically significant differences. At concentrations higher than 3 mg/mL, the antioxidant activity of *Fulvifomes* sp. extract decreased significantly, becoming undetectable at 6 mg/mL.

This behavior deviates from the expected trend that higher concentrations would lead to a constant or greater reduction in DPPH. The observed decrease in activity could be attributed to a more intense brown color at higher concentrations, which could interfere with measurements. Despite adjustments to the sample blank to mitigate this effect, the problem persisted, suggesting the possible formation of aggregates in the test solution. This phenomenon could impede DPPH interaction by affecting the accessibility of the compound, increasing the turbidity of the solution, or overestimating the readings. Similar behavior was observed in the Plus and Tea supplement, where concentrations higher than 0.75 mg/mL showed a loss of linearity of the response, with a marked decrease in the DPPH scavenging capacity. Furthermore, this pattern may indicate an inhibition of antioxidant activity at high doses, possibly due to interference with other antioxidant molecules and a disturbance of the redox balance [25].

#### 2.2.2. ABTS Scavenging Assay

The results of the antioxidant activity of the three aqueous extracts evaluated by the ABTS method are presented in Figure 6a. The extract of *Fulvifomes* sp. showed the highest antioxidant activity, reaching a 74% value. In comparison, the extracts of *Inonotus* sp. and *P. boldo* reached values of 38%, with no statistically significant differences. However, all the extracts showed a lower antioxidant activity than vitamin C, which showed a 97% value.

In Figure 6b, the reference antioxidants, such as vitamin C, trolox, and gallic acid, show a considerably elevated and sustained antioxidant activity. In particular, the curve corresponding to vitamin C is not visible since it overlaps with gallic acid. On the other hand, the extract evaluated shows a decrease in its antioxidant capacity at concentrations higher than 3 mg/mL, exhibiting behavior similar to that of Tea, but with more significant antioxidant activity compared to the Plus supplement.

These results suggest that, although the extracts have a relevant antioxidant profile, the complexity of their matrices affects the total activity, possibly due to interactions between their components that limit the availability of the active antioxidant compounds. The literature supports this phenomenon, indicating that the composition of the extracts, which includes non-antioxidant compounds, such as carbohydrates and lipids, can reduce the accessibility of antioxidants to interact with free radicals [26]. Furthermore, although pure antioxidants such as vitamin C exhibit a high free radical neutralization capacity under in vitro conditions, natural extracts often benefit complex biological systems due to synergistic effects between their additional components [15].

#### 2.2.3. Hydroxyl Radical Scavenging Assay

The results of the hydroxyl radical scavenging assay for the three aqueous extracts are presented in Figure 7a. As in the previous methods, the *Fulvifomes* sp. extract showed the highest antioxidant capacity among the samples tested, with a value of 77%. The *Inonotus* sp. and *P. boldo* extracts presented antioxidant activity values of 36% and did not show significant differences between them. However, all three extracts presented lower antioxidant activity than vitamin C, which reached a value of 98%.

A significant difference in antioxidant capacity is observed when comparing *Fulvifomes* sp. extract with standard antioxidants such as vitamin C, trolox, and gallic acid at various concentrations (Figure 7b). These standards present an antioxidant activity close to 100%, standing out as potent free radical scavengers. This difference in efficacy can be attributed to the nature of the compounds. Vitamin C, trolox, and gallic acid are pure antioxidants, thus showing maximum activity in this type of assay [15]. Furthermore, this method allowed for quantifying the antioxidant capacity at all concentrations evaluated. In this context, the extract showed the highest antioxidant activity compared to the Plus supplement and Tea, which increased with concentration.

Studies have documented that mushroom extracts can exhibit significant antioxidant activities due to phenolic compounds, polysaccharides, and other bioactive metabolites. However, the antioxidant activity of these extracts is often lower than that of pure antioxidants due to the presence of other components in the mixtures. Zhao and collaborators [15] point out that crude extracts may contain sugars and other non-antioxidant compounds that dilute the effectiveness of the active components. Along these lines, previous studies have reported that *Inonotus* and *Phylloporia* extracts exhibit intermediate antioxidant activity compared to other medicinal mushrooms, which is consistent with our results. For example, Wang and collaborators [27] showed that *Inonotus obliquus* extracts exhibit moderate activity in free radical scavenging assays attributed to phenolic compounds, although not as high as pure antioxidants.

These results demonstrate a high antioxidant efficacy of vitamin C in all the methods evaluated, providing a solid reference for comparing the antioxidant activity of the tested extracts. Furthermore, *Fulvifomes* sp. extract showed the best results for scavenging DPPH, ABTS, and H_2_O_2_ radicals compared to the other two extracts, as shown in Figure 5a, Figure 6a, and Figure 7a. On the other hand, this extract showed the best results for all three methods evaluated compared to Tea and the Plus supplement, as seen in Figure 3B, Figure 4b, and Figure 5b.

Each method used to assess antioxidant activity, such as DPPH, ABTS, and H_2_O_2_ scavenging, measures different aspects of antioxidant capacity, influenced by the solubility, affinity, and specific reactivity of the antioxidant compounds, thus providing unique and complementary information on the antioxidant capacity of the compounds evaluated, and are not compared to each other [26].

It is important to note that the presence of antioxidant compounds in extracts does not guarantee activity similar to that of pure antioxidants. The complex matrix of extracts may reduce the accessibility of active compounds to free radicals, affecting their effectiveness [27]. Therefore, these extracts may be more effective in complex biological systems, where additional components may have synergistic effects. However, pure antioxidants or more concentrated extracts may be preferable for applications with high antioxidant activity.

### 2.3. Cytotoxicity

#### 2.3.1. Fibroblast Viability

Cytotoxicity analyses were performed using a fluorescence-based method combining Hoechst 33342, PI, and the pH-sensitive probe BCECF-AM, to evaluate the effects of aqueous extracts of *Inonotus* sp., *Fulvifomes* sp., and *P. boldo* at a 3 mg/mL concentration on fibroblast viability. This approach allows integrated visualization of three key cellular parameters: nuclear localization (blue fluorescence from Hoechst 33342), cell death (red fluorescence from PI), and physiological pH status (green fluorescence from BCECF-AM) [28].

Control cells (untreated) are shown in Figure 8, displaying low PI fluorescence (indicative of low cell death), a well-defined blue nuclei (Hoechst), and a uniform green BCECF-AM signal, consistent with normal cell physiology. These were compared with treated cells shown in Figure 9, Figure 10 and Figure 11 for *Inonotus* sp., *Fulvifomes* sp., and *P. boldo*, respectively. All images were acquired at 10× magnification, allowing a complete view of the 24-well plate fields.

Cells treated with the *Inonotus* sp. extract (Figure 9) did not exhibit marked cell death; however, alterations in BCECF-AM and slight increases in PI signal compared to controls suggest some compromise in membrane integrity and metabolic status. Conversely, cells exposed to the *P. boldo* extract (Figure 11) showed a response more similar to the control, with minimal PI fluorescence and only slight changes in BCECF-AM signal, suggesting a milder impact on cell viability.

The *Fulvifomes* sp. extract (Figure 10) also did not induce significant cell death, but localized changes in BCECF-AM intensity were observed, particularly in denser cell areas. These changes were accompanied by a slight reduction in PI signal, possibly indicating an effect on cell metabolism rather than membrane integrity.

These observations suggest that none of the extracts caused severe cytotoxicity at 3 mg/mL. The consistently low PI fluorescence across all samples indicates preserved membrane integrity and limited cell death. Meanwhile, green BCECF-AM fluorescence supports the presence of viable cells with normal intracellular pH. The appearance of yellow signals in merged images may reflect overlapping signals from live and dead cells or intermediate viability states but does not point to significant cytotoxicity. Among the three, *Fulvifomes* sp. showed the least physiological disturbance, while *Inonotus* sp. and *P. boldo* caused more noticeable—though still moderate—effects on cell physiology.

#### 2.3.2. WST-1 Cell Proliferation Assay

The toxicity of each aqueous extract was evaluated in the RTgill-W1 salmonid cell line by the WST-1 Cell Proliferation Assay method to 3 and 6 mg/mL, the results of which are shown in Figure 12. A significant decrease in cell viability was observed at the highest concentration of 6 mg/mL compared to the negative control. *Inonotus* sp. presented a cell viability of 79%, suggesting low cytotoxicity and relatively safe activity compared to the other extracts. In contrast, *Fulvifomes* sp. and *P. boldo* showed markedly lower viabilities (42%), with no significant difference between them, suggesting higher cytotoxicity.

These results indicate that *Inonotus* sp. is the least toxic extract at 6 mg/mL, potentially due to a bioactive profile that does not compromise cell integrity at this concentration. The more significant cytotoxicity observed for *Fulvifomes* sp. and *P. boldo* may be associated with their higher content of phenolic compounds, which are known to act as antioxidants at low concentrations but can induce oxidative stress and reduce viability at higher doses [25,29].

When the concentration was reduced to 3 mg/mL, matching that used in the antioxidant assays, cell viability increased across all samples. *Inonotus* sp. reached 90% viability, approaching that of the negative control and reaffirming its low toxicity and good compatibility with the salmonid cell line. *Fulvifomes* sp. showed improved viability at this concentration (65%), indicating a reduction in cytotoxicity, while *P. boldo* maintained a similar viability to that observed at 6 mg/mL (44%), suggesting a plateau effect.

These findings highlight *Inonotus* sp. as the safest extract among those tested, demonstrating minimal cytotoxicity at both concentrations evaluated. Conversely, *Fulvifomes* sp. and *P. boldo* maintained lower viability levels, which suggests that, despite reduced toxicity at 3 mg/mL, certain compounds within these extracts still elicit cytotoxic effects in RTgill-W1 cells. This is consistent with previous reports indicating that *P. boldo* extracts exert cytotoxicity due to high levels of phenolic compounds and terpenoids [30].

The observed cytotoxic profiles align with earlier studies showing that the biological effects of fungal extracts vary depending on concentration and the specific composition of bioactive molecules [31,32]. In this context, *Inonotus* sp. demonstrated the most favorable profile, supporting its potential for applications that require low cytotoxicity, such as aquaculture health strategies or antioxidant supplement development. Due to their higher cytotoxic potential, *Fulvifomes* sp. and *P. boldo* should be used cautiously in similar applications.

## 3. Materials and Methods

### 3.1. Materials and Chemical

The fungal material used in this study consists of basidiomata collected from endemic trees in various localities of the Valparaíso region. In situ, a portion of the basidioma (internal part) measuring approximately 125 cm^3^ (5 cm × 5 cm × 5 cm) was extracted directly from the host tree using a field knife. The samples were collected and stored in paper envelopes, then transported to the laboratory, where they were dehydrated in a NEX food dehydrator at 35 °C for 72 h. Afterward, they were preserved in cellophane bags and stored in sealed plastic containers. A 20 g sample of each fungal species was taken from the dehydrated basidiomata for subsequent analyses. *Fulvifomes* sp. was collected from *Schinus latifolius* in Cartagena, San Antonio province, at coordinates 33°32′10″ S, 71°36′42″ W, at 96 m above sea level (m.a.s.l.), in October 2018. *P. boldo* was collected from *Peumus boldus* in the same locality of Cartagena, at coordinates 33°32′08″ S, 71°36′46″ W, at 90 m.a.s.l., in October 2022. *I.* aff. *quercustris* was collected from *Quillaja saponaria* in the locality of Limache, Marga Marga province, at coordinates 33°00′39″ S, 71°15′56″ W, at 566 m.a.s.l., during August 2022. The RTgill-W1 ATCC CRL-2523 cellular line was acquired from the American Type Culture Collection (ATCC). Analytical-grade reagents were used in all experiments.

The morphological identification of basidiomata was performed based on the global literature on poroid *Hymenochaetaceae*, utilizing taxonomic keys and detailed descriptions to determine the genus or species level in each case [33,34]. This was subsequently complemented with molecular characterization using the ITS (Internal Transcribed Spacer) region. DNA was extracted using a CTAB buffer protocol modified by Ossa et al. [35]. The ITS1 and ITS2 regions were amplified using the primers ITS1 (5′ TCCGTAGGTGAACCTGCGG 3′) and ITS4 (5′ TCCTCCGCTTATTGATATGC 3′) as described by White et al. [36]. Successful amplification was confirmed via electrophoresis on a 2% agarose gel, and the PCR products were sent for Sanger sequencing. The resulting sequences were edited, aligned, and assembled into consensus sequences using BioEdit v7.2.5 [37].

Phylogenetic analysis was performed by comparing the obtained sequences with closely related taxa in the NCBI BLAST database (https://blast.ncbi.nlm.nih.gov/Blast.cgi, accessed on 20 January 2024). Sequences from the genera *Inonotus*, *Fulvifomes*, and *Phylloporia* were downloaded from GenBank and aligned with the study sequences. Maximum Likelihood analysis was conducted using IQ-TREE v2.0 [38], applying the best-fit substitution model (HKY + F + I + G4) determined according to the Bayesian Information Criterion (BIC). The resulting phylogenetic tree was visualized using FigTree v1.4.5 [39] and is presented in Figure A1.

Precise taxonomic identification of *Inonotus aff. quercustris* and *Fulvifomes* sp. remains under development due to the morphological complexity and limited molecular data available for species endemic to the Chilean sclerophyllous forest. The sequence obtained for *Inonotus aff. quercustris* has been deposited in GenBank under accession number [PV007713.1], and it may represent a previously undescribed species according to expert evaluation. For this reason, it is provisionally referred to as *Inonotus* sp. in this manuscript. The sequence of *Fulvifomes* sp. has also been submitted and is currently under phylogenetic analysis (GenBank code: [PV459630.1]). These limitations have been considered when interpreting the results and are further addressed in the discussion section.

### 3.2. Preparation of Fungal Aqueous Extracts

Aqueous extraction of fungi was performed according to Hwang et al. [40], with some modifications. For this, 6 g of dried fungal material was weighed and previously ground (using a disinfected electric blade grinder (Thomas, Neunkirchen, Germany)) until a homogeneous powder was obtained for each species (*Fulvifomes* sp., *Inonotus* sp., and *P. boldo*). The resulting powders were transferred to 250 mL beakers, to which 60 mL of distilled water was added. The mixtures were heated to 98 °C with constant stirring for 1 h. The suspensions were then subjected to a 10 min sonication bath to facilitate the release of bioactive compounds. The samples (60 mL) were centrifuged at 2000× *g* at 25 °C for 10 min in a centrifuge (Hettich Rotina 380, Tuttlingen, Germany) to separate the supernatant from solid residues. Finally, the supernatant was recovered and lyophilized, and the lyophilized extract was stored at 4 °C in the dark for further testing. This aqueous extraction method was chosen over organic solvents due to its advantages in retaining polar bioactive compounds like polysaccharides, proteins, and certain phenolic compounds, which contribute significantly to the antioxidant properties of fungi extracts. Furthermore, aqueous extraction ensures non-toxicity, aligns with traditional medicinal preparations, and is compatible with direct applications in biological assays without the risk of residual solvent toxicity [41], making it an ideal choice for bioactivity studies in this context.

### 3.3. Characterization of Aqueous Extracts

#### 3.3.1. Determination of Total Phenolic Compounds

For the quantification of polyphenols, a calibration curve was prepared using gallic acid at concentrations of 10, 50, 120, 200, 300, and 400 µg/mL, according to Singleton and Rossi [42]. In the analysis procedure, 50 µL of the sample under study was added to a test tube, together with 50 µL of distilled water, with 10 µL of Folin–Ciocalteu reagent diluted in a ratio of 1:3 and 40 µL of 20% (*w*/*v*) sodium carbonate solution. Subsequently, the mixture was completed with an additional 50 µL of distilled water. To allow the reaction to stabilize, the mixture was allowed to stand for 30 min at 25 °C. Finally, the absorbance at 750 nm was measured using a microplate reader (VersaMax, San Jose, CA, USA).

#### 3.3.2. Determination of Total Protein

The protein quantification by the Lowry method was realized according to Fryer et al. [43], with some modifications. A calibration curve was prepared using bovine serum albumin (BSA) at 50, 100, 200, 320, and 500 µg/mL concentrations. The alkaline carbonate–tartrate–copper (CTC) solution was prepared by dissolving copper sulfate pentahydrate (CuSO_4_·5H_2_O) at 0.1% (*w*/*v*), sodium tartrate at 0.2% (*w*/*v*), and sodium carbonate (Na_2_CO_3_) at 10% (*w*/*v*) in water. The alkaline copper reagent (ACR) was prepared by mixing the CTC solution with 0.8 M NaOH and water in a 1:1:2 ratio. For the quantification, 50 µL of the sample was to be analyzed, and 100 µL of the ACR was added and left to react for 10 min. Then, 50 µL of Folin–Ciocalteu reagent diluted in water at a 1:5 ratio was incorporated. The mixture was left to react for 30 min at 25 °C before measuring the absorbance at 620 nm using a microplate reader (VersaMax, San Jose, CA, USA).

#### 3.3.3. Identification of Phenolic Compounds

The composition of the polyphenol profile in the aqueous extracts of the fungi was analyzed by HPLC-FLD, according to the method described by Elena Tarnok et al. [44]. For these analyses, an UltiMate 3000 HPLC system (Thermo Fisher Scientific Inc., Waltham, MA, USA) equipped with an LPG-3400 SD quaternary gradient pump, a TCC 3000SD column oven, and an FLD3100SD fluorescence detector was used. A ZORBAX Eclipse Plus C18 column (4.6 × 150 nm, particles: 5 μm) purchased from Agilent Technologies (Santa Clara, CA, USA) was used for separation. The separation was conducted in gradient mode. Chromatographic analyses were performed at 30 °C with a 1 mL/min flow rate. The mobile phases used were acetonitrile and 5% (*v*/*v*) acetic acid. A mixture of both phases was used for the run, performing a gradient between 9% and 30% acetonitrile from 0 to 20 min, then a gradient between 30% and 85% acetonitrile between 20 and 23 min. An excitation wavelength of 280 nm was used to perform the chromatographic profile scans, and the emission was measured between 320 nm and 520 nm. The extract samples were prepared at a concentration of 1 mg/mL.

### 3.4. Antioxidant Activity

#### 3.4.1. DPPH Scavenging Assay

The antioxidant capacity of aqueous extracts of *Fulvifomes* sp., *Inonotus* sp., and *P. boldo* were evaluated at a concentration of 3 mg/mL, using vitamin C as a positive control. The 2,2-diphenyl-1-picrylhydrazyl (DPPH) radical scavenging analysis followed the methodology described by Nuñez et al. [45], with some modifications. In a 96-well microplate, 18 µL of the sample was added with 182 µL of a DPPH solution at 30 mg/L. For the blank, 18 µL of the sample was combined with 182 µL of distilled water, while 18 µL of distilled water and 182 µL of the DPPH solution were used as a control. The microplate was incubated in the dark and continuously stirred for 30 min to prevent photodegradation of the radical. After this incubation period, the absorbance was measured at 517 nm using a microplate reader (VersaMax, San Jose, CA, USA). The scavenging activity on DPPH radical was calculated from the absorbances obtained, as shown in Equation (1), which provided a measure of the antioxidant capacity of the samples evaluated.(1)$$Scavenging\;activity\;\left(\%\right)=\left[\frac{Abs\;control-(Abs\;sample-Abs\;blank)}{Abs\;control}\right]\times 100,$$
where *Abs sample* is the absorbance of extracts with DPPH, *Abs control* is the absorbance of DPPH without any extract, and *Abs blank* represents the absorbance of extracts without DPPH. The DPPH with ethanol was used as a control, and water with extracts was blank.

#### 3.4.2. ABTS Scavenging Assay

The antioxidant capacity of aqueous extracts of *Fulvifomes* sp., *Inonotus* sp., and *P. boldo* were evaluated at a concentration of 3 mg/mL, using vitamin C as a positive control. The 2,2′-Azino-Bis(3-Ethylbenzthiazoline-6-Sulphonic Acid (ABTS) radical scavenging analysis was performed following the methodology described by Costamagna et al. [46], with some modifications. Solutions of ABTS at a concentration of 7 mM and potassium persulfate at 2.45 mM were prepared, mixed in a 1:1 ratio, and allowed to stand for 14 h to facilitate the formation of the ABTS radical. The resulting solution was diluted in ethanol (99.99%) to reach an absorbance of approximately 0.7 at 734 nm; this was accomplished by performing a 1:50 dilution. In the microplate, 10 µL of the sample was added to 190 µL of the diluted ABTS solution. For the control, 10 µL of ethanol and 190 µL of the ABTS solution were used. The mixture was incubated and continuously stirred for 6 min at room temperature, and the absorbance was immediately measured at 734 nm. For the colored samples, a blank consisting of 10 µL of the sample and 190 µL of ethanol was used. The percentage of free radical capture was calculated from the measured absorbances, as indicated in Equation (2).(2)$$\;Scavenging\;activity\;\left(\%\right)=\frac{\left(Abs\;blank-Abs\;sample\right)}{Abs\;blank}\times 100,$$
where *Abs sample* is the absorbance of the sample, and *Abs blank* is the absorbance of the blank.

#### 3.4.3. Hydroxyl Radical Scavenging Assay

Hydroxyl radical scavenging activity was determined following the method described by Mu et al. [6], with minor modifications for a 96-well microplate format. Briefly, 13 µL of the sample (extract), 13 µL of FeSO_4_ (9.1 mM), 80 µL of distilled water, and 13 µL of salicylic acid (9.1 mM) were added to each well. The reaction was initiated by adding 81 µL of H_2_O_2_ (8.8 mM). Absorbance was recorded at 510 nm to evaluate the hydroxyl radical scavenging capacity. For the control, H_2_O_2_ was replaced with distilled water, and for the blank, distilled water was added instead of the sample. The scavenging percentage was calculated using Equation (3), which reflects the ability of the sample to scavenge hydroxyl radicals.(3)$$Scavenging\;activity\;\left(\%\right)=1-\frac{\;(Abs\;sample-Abs\;control)}{Abs\;blank}\times 100,$$

### 3.5. Cytotoxicity

#### 3.5.1. Fibroblast Viability

Cytotoxicity was assessed by evaluating fibroblast viability at a 3 mg/mL concentration for each extract using a fluorescence-based assay with multiple fluorochromes. Propidium iodide (PI), which penetrates cells only when the plasma membrane is compromised, was used to detect non-viable cells through red fluorescence. Hoechst 33342 dye, which binds preferentially to adenine–thymine-rich regions in the minor groove of DNA, was used to stain cell nuclei, emitting blue fluorescence [47]. Additionally, the pH-sensitive probe BCECF-AM (2′,7′-bis-(2-carboxyethyl)-5-(and-6)-carboxyfluorescein acetoxymethyl ester) was included to evaluate intracellular pH as an indicator of cell physiology [48].

The assay was conducted in a 24-well microplate with a final volume of 500 µL per well. Fibroblast cells were seeded at a 20,000 cells/cm² density and grown to 100% confluence. Cells were washed twice with pre-warmed Krebs–Henseleit (KH) (118 mM NaCl, 4.7 mM KCl, 1.25 mM CaCl_2_, 1.2 mM MgSO_4_, 1.2 mM KH_2_PO_4_, 25 mM NaHCO_3_) buffer supplemented with 0.5 mM Ca^2+^ and 5 mM lactate, followed by a 1 min incubation in the same buffer at 37 °C. BCECF-AM was then added and incubated for 30 min at 37 °C. After incubation, the medium was removed and replaced with a staining solution containing Hoechst 33342 and PI, followed by a 10 min incubation at 37 °C. Two final washes were performed with KH buffer before microscopic observation.

A fluorescence microscope (Nikon Ti-E, Tokyo, Japan) was used for image capture and equipped with a coupled digital camera system for image acquisition. Fluorescence detection was carried out using the following filter sets: for Hoechst 33342 (blu), excitation at 350 nm, dichroic mirror at 400 nm, emission at 460 nm, and 0.5 s exposure; for PI (red), excitation at 530 nm, dichroic mirror at 570 nm, emission at 630 nm, and 1 sec exposure; and for BCECF-AM, excitation at 485 nm, dichroic mirror at 510 nm, emission at 535 nm, and 1 sec exposure. To assess basal viability, the control used for comparison consisted of cells not treated with extracts, exposed only to KH buffer and fluorochromes.

For the interpretations, blue nuclei (Hoechst 33342) indicated cells with accessible DNA. Combined with the absence of a red signal (PI), they confirm that the cell membrane remains intact. The green signal (BCECF-AM) reflects a normal physiological pH.

Non-viable cells were identified by red-stained nuclei (PI), indicating a compromised membrane allowing fluorochrome entry. An increased red signal (PI) in treated samples compared to the negative control suggests cytotoxicity, while changes in BCECF-AM fluorescence indicate metabolic or pH alterations.

Comparison with the negative control establishes the baseline for viability and metabolism, allowing us to distinguish treatment-specific effects from experimental variability. This approach enabled the simultaneous evaluation of cell viability and physiological status, allowing clear differentiation between viable and non-viable cells and providing a detailed assessment of the cytotoxic effects of each extract.

#### 3.5.2. WST-1 Cell Proliferation Assay

Additionally, the toxicity of the extracts was evaluated in the RTgill-W1 salmonid cell line by the WST-1 Cell Proliferation Assay method [49], a colorimetric assay that measures mitochondrial dehydrogenase activity. The RT-gill cell line was prepared by washing with phosphate-buffered saline (PBS) 1×. When the necessary confluence for the assay was reached, trypsinization was performed to detach the cells from the flask. The cell suspension was transferred to a Falcon tube and centrifuged at 180× *g* for 5 min; finally, the pellet was resuspended in L-15 medium, and the cells were counted in a Neubauer chamber using Trypan Blue. The necessary dilution was made with culture medium to achieve a concentration of 7.0 × 10^4^ viable cells/mL, and 100 µL of the cell suspension was added to each well of a 96-well plate and incubated for 24 h at 18 °C until reaching 80% confluence, ideally forming a monolayer. The extracts to be tested were then added at concentrations of 3 and 6 mg/mL, and after 48 h of exposure, the WST-1 reagent was added. A 20% Triton X-100 solution in a medium was used as a positive control. The WST-1 cleavage product was measured after 4 h at 450 nm. WST-1 plus medium was used as a blank (without cells), which was subtracted from all values. Cell viability percentages were calculated using Equation (4).(4)$$Cell\;viability\;\left(\%\right)=\frac{\;Abs\;sample}{Abs\;control}\times 100$$
where *Abs control* is the absorbance of the negative control (untreated cells), and *Abs sample* is the absorbance of the evaluated extracts.

### 3.6. Statistical Analysis

The data are presented as the mean value ± standard deviation of three replicates. The data were subjected to the Shapiro–Wilk normality test and ANOVA with a Tukey multiple comparison for antioxidant activity. Statistical analyses were performed in GraphPad Prism, version 10 for Windows, GraphPad Software, Boston, MA, USA.

## 4. Conclusions

This study evaluated the antioxidant activity and cytotoxicity of aqueous extracts from three fungal species: *Fulvifomes* sp., *P. boldo*, and *Inonotus* sp. Among the three, *Fulvifomes* sp. showed the highest polyphenol content and the strongest antioxidant capacity across DPPH, ABTS, and hydroxyl radical scavenging assays. Conversely, *P. boldo* exhibited a higher protein concentration but lower antioxidant activity. In terms of cytotoxicity, *Inonotus* sp. demonstrated the lowest toxic effect on RTgill-W1 cells, particularly at 3 mg/mL, supporting its potential as a safe bioactive agent. The results suggest that *Inonotus* sp. could be considered for use in aquaculture and functional applications, while the *Fulvifomes* sp. and *P. boldo* extracts require caution at higher concentrations due to their cytotoxic profiles. Overall, the study underscores the importance of native fungi from sclerophyll forests as promising but underexplored sources of natural antioxidants, advocating for continued research on extraction optimization, compound isolation, and targeted applications in the food, pharmaceutical, and aquaculture sectors. Although there are limitations in the precise taxonomic identification of some samples, the results remain relevant and promising, opening new opportunities to explore the bioactive potential of little-studied fungal species from Chilean sclerophyllous forests.

## Figures and Tables

**Figure 1 ijms-26-05877-f001:**
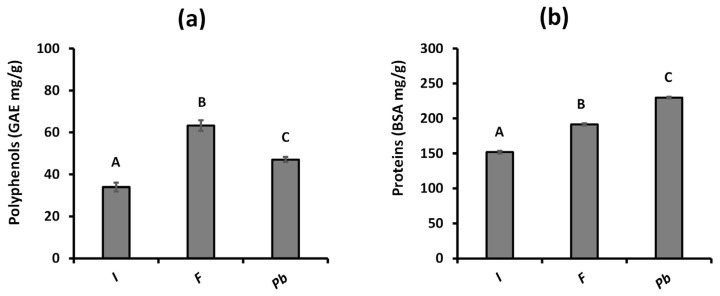
Concentration of total polyphenols (**a**) and protein (**b**) in aqueous extracts of *Inonotus* sp. (I), *Fulvifomes* sp. (F), and *Phylloporia boldo* (Pb). Values are the mean of three replicates ± standard deviation. Different letters indicate significant differences (*p* ≤ 0.0001) among all groups.

**Figure 2 ijms-26-05877-f002:**
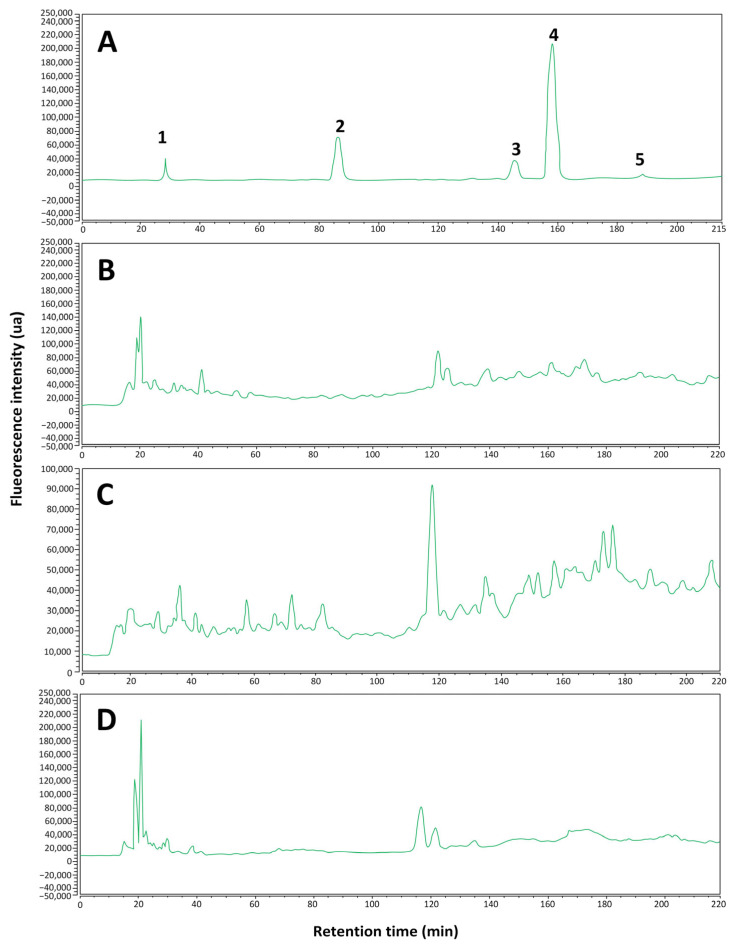
Chromatograms with emission signal obtained at 360 nm of the standard compounds (**A**) and of the aqueous extracts of *P. boldo* (**B**), *Fulvifomes* sp. (**C**), and *Inonotus* sp. (**D**). The labeled peaks for the chromatogram of the standards are 1. gallic acid, 2. catechin, 3. epicatechin, 4. syringic acid, and 5. *p*-coumaric acid.

**Figure 3 ijms-26-05877-f003:**
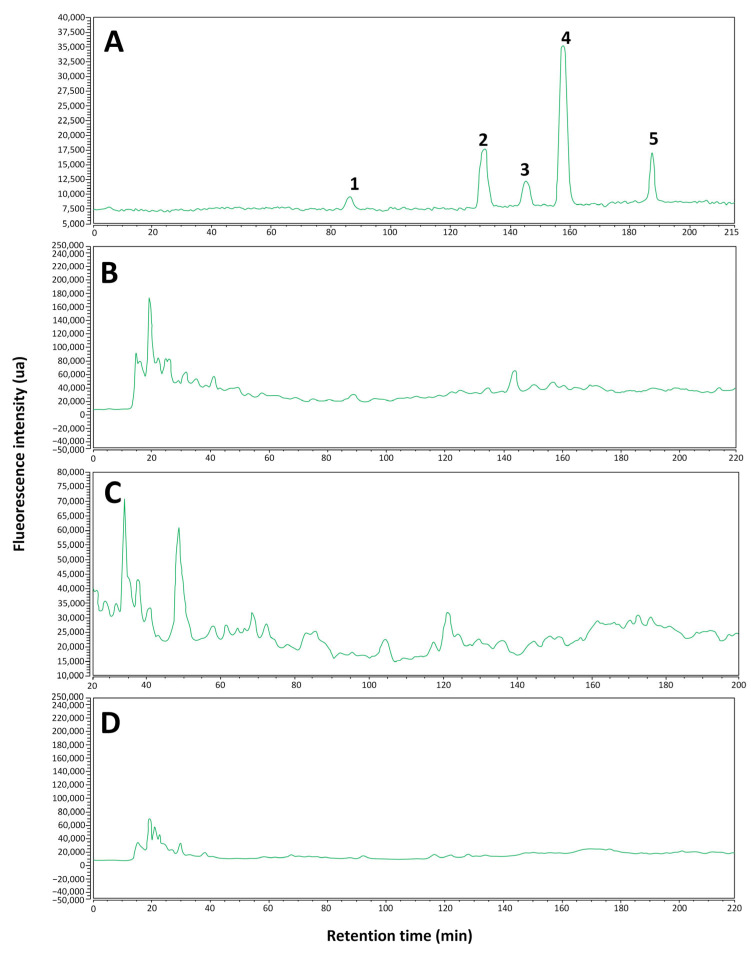
Chromatograms with emission signal obtained at 422 nm of the standard compounds (**A**) and of the aqueous extracts of *P. boldo* (**B**), *Fulvifomes* sp. (**C**), and *Inonotus* sp. (**D**). The labeled peaks for the chromatogram of the standards are 1. catechin, 2. caffeic acid, 3. epicatechin, 4. syringic acid, and 5. *p*-coumaric acid.

**Figure 4 ijms-26-05877-f004:**
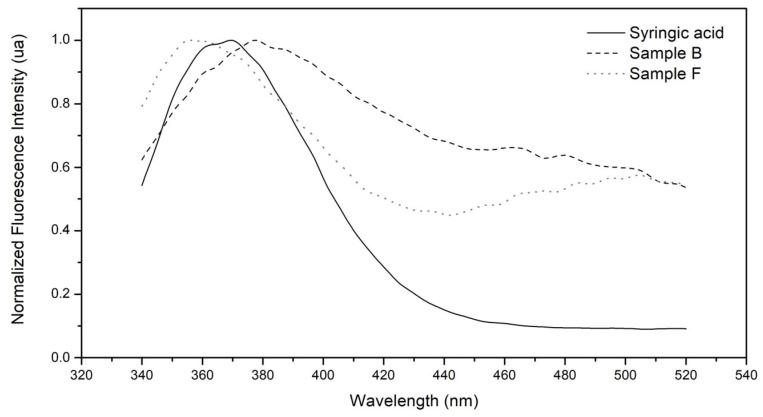
Fluorescence emission spectrum obtained from the retention time chromatograms of syringic acid. The solid line spectrum is the one extracted from the chromatogram of the syringic acid standard, the one labeled as sample B is the equivalent for the aqueous extract of *P. boldo*, and sample F is for the extract of *Fulvifomes* sp.

**Figure 5 ijms-26-05877-f005:**
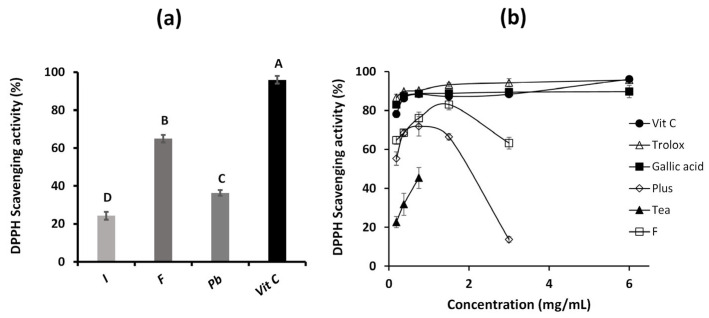
DPPH scavenging activity of: *Inonotus* sp. (I), *Fulvifomes* sp. (F), *Phylloporia boldo* (Pb), extracts of vitamin C (Vit C) assayed at 3 mg/mL (**a**,**b**) *Fulvifomes* sp. extract, Vit C, Trolox, Gallic acid, black tea with blueberries and maqui (Tea), and the collagen supplement fortified with Vit C and magnesium (Plus) tested at different concentrations. Values are the mean of three replicates ± standard deviation. Different letters indicate significant differences (*p* ≤ 0.0001) among all groups.

**Figure 6 ijms-26-05877-f006:**
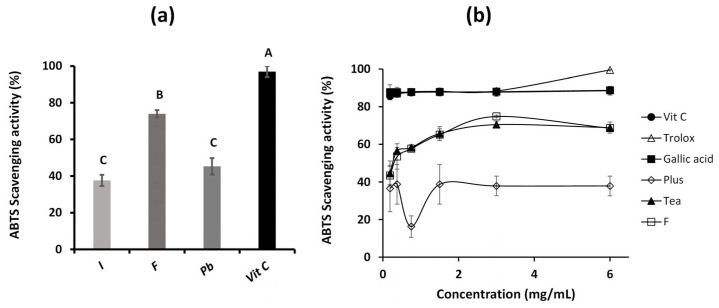
ABTS scavenging activity of: *Inonotus* sp. (I), *Fulvifomes* sp. (F), *Phylloporia boldo* (Pb), extracts of vitamin C (Vit C) assayed at 3 mg/mL (**a**,**b**) *Fulvifomes* sp. extract, Vit C, Trolox, Gallic acid, black tea with blueberries and maqui (Tea), and the collagen supplement fortified with Vit C and magnesium (Plus) tested at different concentrations. Values are the mean of three replicates ± standard deviation. Different letters indicate significant differences (*p* ≤ 0.0001) among all groups.

**Figure 7 ijms-26-05877-f007:**
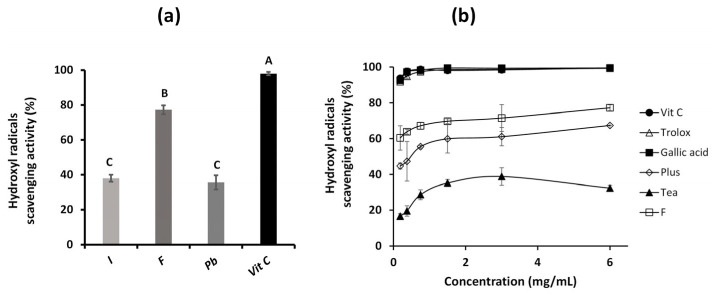
Scavenging activity on hydroxyl radicals of: *Inonotus* sp. (I), *Fulvifomes* sp. (F), *Phylloporia boldo* (Pb), extracts of vitamin C (Vit C) assayed at 3 mg/mL (**a**,**b**) *Fulvifomes* sp. extract, Vit C, Trolox, Gallic acid, black tea with blueberries and maqui (Tea) and the collagen supplement fortified with Vit C and magnesium (Plus) tested at different concentrations. Values are the mean of three replicates ± standard deviation. Different letters indicate significant differences (*p* ≤ 0.0001) among all groups.

**Figure 8 ijms-26-05877-f008:**
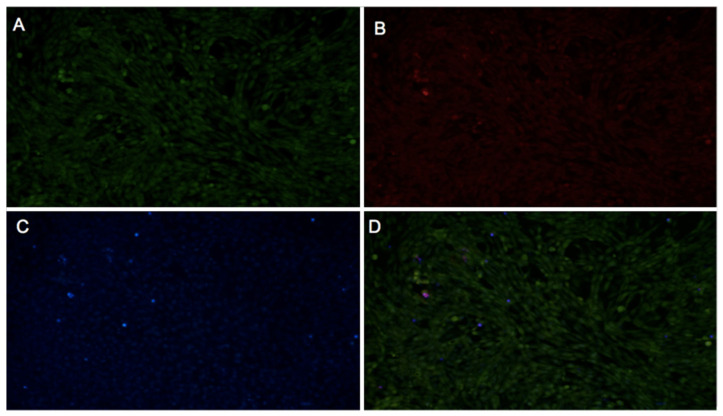
Fluorescent microscopy images of fibroblast without extract (control sample) and stained with BCECF-AM (**A**), propidium iodide (**B**), Hoechst 33342 (**C**), or overlap (**D**) stains. Three separate pictures from the same field were taken for the three markers. Scale bar = 50 μm.

**Figure 9 ijms-26-05877-f009:**
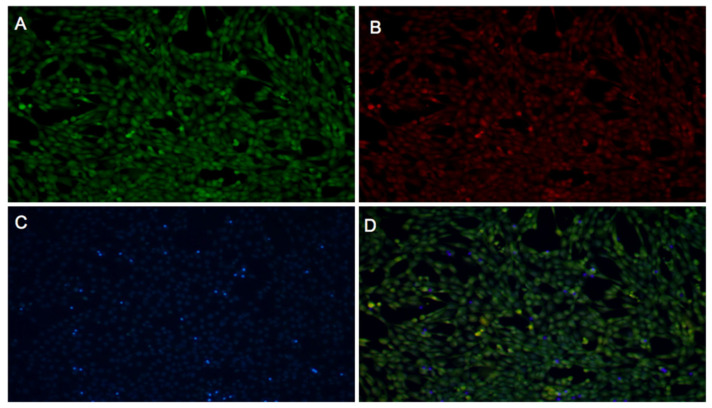
Fluorescent microscopy images of fibroblast treated with aqueous extract (3 mg/mL) of *Inonotus* sp. and stained with BCECF-AM (**A**), propidium iodide (**B**), Hoechst 33342 (**C**), or overlap (**D**) stains. Three separate pictures from the same field were taken for the three markers. Scale bar = 50 μm.

**Figure 10 ijms-26-05877-f010:**
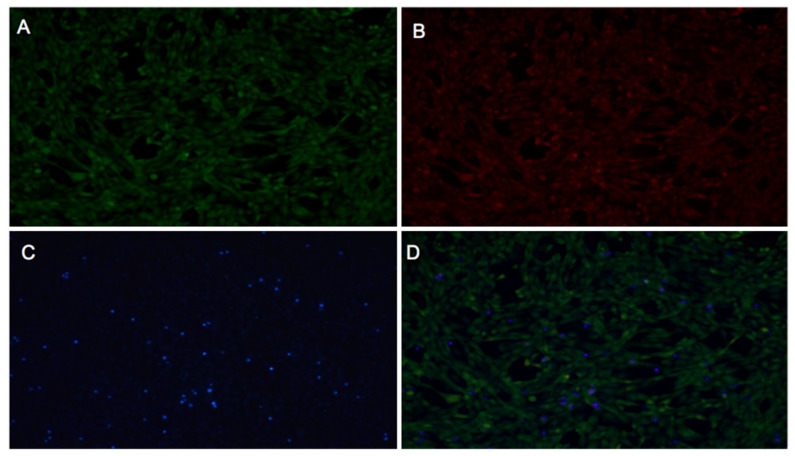
Fluorescent microscopy images of fibroblast treated with aqueous extract (3 mg/mL) of *Fulviformes* spp. and stained with BCECF-AM (**A**), propidium iodide (**B**), Hoechst 33342 (**C**), or overlap (**D**) stains. Three separate pictures from the same field were taken for the three markers. Scale bar = 50 μm.

**Figure 11 ijms-26-05877-f011:**
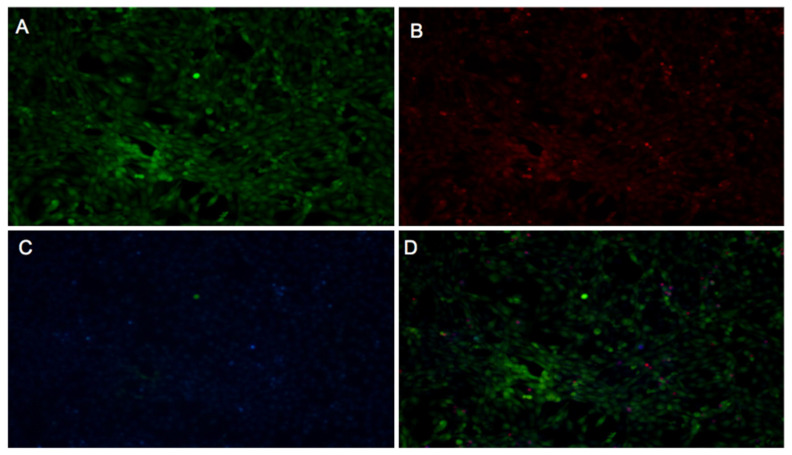
Fluorescent microscopy images of fibroblast treated with aqueous extract (3 mg/mL) of *P. boldo* and stained with BCECF-AM (**A**), propidium iodide (**B**), Hoechst 33342 (**C**), or overlap (**D**) stains. Three separate pictures from the same field were taken for the three markers. Scale bar = 50 μm.

**Figure 12 ijms-26-05877-f012:**
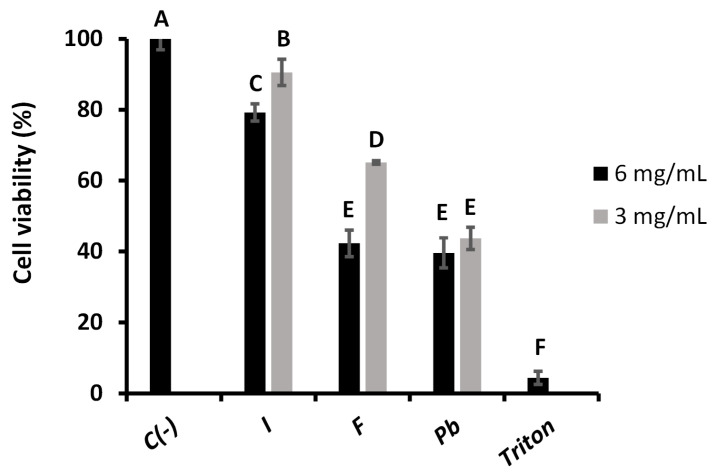
Viability of RTgill-W1 salmonid cell line evaluated by WST-1 cell proliferation assay after treatments with two different concentrations of *Inonotus* sp. (I), *Fulvifomes* sp. (F), or *Phylloporia boldo* (Pb) aqueous extract. Untreated cells (C(-)) and Triton (T) were performed as negative and positive control, respectively. Values are the mean of three replicates ± standard deviation. Different letters indicate significant differences (*p* ≤ 0.0001) among all groups.

**Table 1 ijms-26-05877-t001:** Identification of phenolic compounds in aqueous fungal extracts by HPLC-FLD.

Polyphenolic Compounds	Presence of Polyphenols in Aqueous Extract		
	*Inonotus* sp.	*Fulvifomes* sp.	*P. boldo*
Gallic acid	no	no	no
Catechin	no	no	no
Epicatechin	no	no	no
Syringic acid	no	yes	yes
Caffeic acid	no	no	no
*p*-Coumaric acid	no	no	no

## Data Availability

The data presented in this study are available on request from the corresponding author due to privacy or ethical restrictions.

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
