# Peer review of "Antioxidant and Cytotoxic Evaluation of Aqueous Extracts from Hymenochaetaceae Fungi Associated with Endemic Chilean Sclerophyll Forest Trees"

_ijms, 2025, doi:10.3390/ijms26125877_

Round 1

Reviewer 1 Report

Comments and Suggestions for Authors

Dear Authors,

Your manuscript is well written. In the attached document, you will find just a few minor suggestions.

Author Response

Response to Reviewers and Editor:

Manuscript Ref: ijms-3680680

The authors would like to thank the editor, and the reviewers for thoroughly assessing the manuscript. We have addressed all comments, and additions are marked in red font in the manuscript version that includes the changes.

Response to Reviewer #1’s Comments

Line 52: This shoul be italic

We have reviewed and corrected all the observations, line 53.

Line 61: MAybe add short for "species affinis"

We have corrected it, line 62.

Line 63: It doesn't seem same font size, remove dot between the words

We have corrected it.

Line 81: According to Index Fungorum, current name is Phellinus fastuosus

We have corrected it.

Line 86: Remove dot

We have corrected it.

Line 89: Maybe replace with tree, since wood is generally considered as material and hence dead.

We have corrected it.

Line 111: cellophane or paper bags? or cellophane-paper bags?

We have corrected by cellophane bags.

Line 115: I don't think this is common way of writing UTM coordinates. Maybe also convert to more commonly used Degrees Minutes Seconds coordinate system

The coordinates have been converted to the Degrees Minutes Seconds (DMS) format, which is more widely used and recognized in scientific publications (lines 432-437).

Line 123: How? According to which literature?

We have added the overview and reference, lines 440-443.

Line 138: Not italic

We have corrected it.

Line 155: Volume of the samples?

We have added the sample volume to be centrifuged, line 472.

Line 169: remove comma

The comma was removed.

Line 173: use unbreakable space (ctrl+shift+space) to avoid separating value and unit

We have corrected it.

Line 174: VersaMax

We have corrected it.

Line 184: with what?

The sample is diluted in water, and we've added it to the manuscript.

Line 186: VersaMax

We have corrected it.

Line 193: sign for multiplication, not letter x

We have corrected it.

Line 193: https://www.agilent.com/cs/library/specifications/public/820114-002.pdf

not 5 μm?

We have corrected it.

Line 195: sometimes you use ml and other times mL, uniform

We have uniformed it throughout the manuscript.

Line 207 and 208: unbreakable space

We have corrected it.

Line 213: VersaMax

We have corrected it.

Equation (1), (2), (3) and (4): use sign for multiplication

We have corrected it.

Line 265: sometimes you write with and sometimes without space, make it uniform  

We have uniformed it with space.

Line 269: Japan

We have corrected it.

Line 324: delete space

We have corrected it.

Line 337: add standard deviations

We have added the standard deviation.

Line 351: Since you are always using the scientific names of fungi, I suggest to replace this names with scientific ones.

We have added the scientific names, line 141.

Line 361: italic

We have corrected it.

Figure 2 an 3: I suggest adding the titles of both axis, and if possible increasing the font size.

We have added the title to the axes.

Line 404: In Figure 2 names of the standards are capitalized.

We have normalized it.

Line 430 and 431: What do you mean by this?

Thank you for your observation. To clarify, “Tea” refers to a commercially available black tea beverage infused with blueberries and maqui berries, which is marketed for its antioxidant properties. “Plus” corresponds to a commercial collagen supplement enriched with vitamin C and magnesium, also promoted for health and antioxidant benefits. These products were selected as real-world comparators to evaluate the relative antioxidant potential of the fungal extract in a more applied context.

Line 435: before you wrote with small letter, here is capitalized

We have normalized it.

Line 458, 486, 509: missing comma

We have added the comma, line 247, 275, 298.

Line 487, 509: italic

We have corrected it.

Line 586: unbreakable space

We have corrected it.

Furthermore, an English-speaking research team member corrected English grammar and sentence structure throughout the manuscript.

Reviewer 2 Report

Comments and Suggestions for Authors

Review of ijms-3680680-v1

The study evaluated the antioxidant activity and cytotoxicity of aqueous extracts from three fungal species: Fulvifomes sp., P. boldo, and Inonotus sp. The polyphenol and protein levels were also evaluated. The obtained results suggest that Inonotus sp. could be considered for use in aquaculture and functional applications, while the Fulvifomes sp. and P. boldo extracts require caution at higher concentrations due to their cytotoxic profiles. The obtained findings highlighted the potential of Inonotus sp. as a natural antioxidant with low cytotoxicity and encourages further exploration of native forest fungi as sources of functional bioactive compounds for food, nutraceutical, or aquaculture applications.

Regarding the article, I have next comments and recommendations:

  • Be careful about Latin names of fungal species throughout the article, they should be written in Italics, you have used it but throughout the article sometimes not, and this should be modified, like L. 52-53: Hymenochaetaceae family; L. 361: Hymenochaetaceae family; L. 642: the Fulvifomes and P. boldo extracts; L. 676: the Inonotus clade. Likewise: L. 362-363, in capture of Figure 2: p-coumaric acid – p-should be in Italics; in Table 1 and capture of Figure 1: better: p-Coumaric acid; in capture of Figure 6 and 7: vitamin C and Fulvifomes sp.
  • 86:… derived from Peumus boldus have been attributed…
  • Units: between value and °C should be a space, like 35 °C, 98 °C, 25 °C, 4 °C, etc. Correct throughout the article. L. 155: at 2500 rpm at 25 °C: convert rpm to g (rcf). L. 182: better: 0.8 M NaOH. You use for µg/ml, mg/mL and µL, etc., decide and unify only one way l or L throughout the article.
  • In used equipment complete where is needed: producer, town and country: NEX food dehydrator; electric blade grinder (Thomas, town, Germany); a centrifuge (Hettich Rotina 380, town, Germany); a microplate reader (VERSA Max, town or state, United States); a microplate reader (VERSA max, town or state, USA); a microplate reader (VERSA Max, town or state. USA); a fluorescence microscope (Nikon Ti-E, town, Japan).
  • In Equation 3: use English instead of Spanish: Abs sample and Abs blank.
  • In References you used a mixture of full journal titles and abbreviations. So, in References substitute journal full titles with their abbreviations where it is needed: Redox Biol.; For. Ecol. Manag.; Int. J. Biol. Macromol.; J. Tradit. Complement. Med.; J. Bioeng.; J. Environ. Sci. Health; Mol. Biol. Evol.; Trends Biotechnol.; Anal. Biochem.; Microchem. J.; Food Res. Int.; Crit. Rev. Food Sci. Nutr.; Food Chem.; Waste Biomass Valori; Prep. Bioch. Biotechnol.; Oxidative Med. Cell. Longev.; Nat. Prod. Res.

Author Response

Response to Reviewers and Editor:

Manuscript Ref: ijms-3680680

The authors would like to thank the editor, and the reviewers for thoroughly assessing the manuscript. We have addressed all comments, and additions are marked in red font in the manuscript version that includes the changes.

Response to Reviewer #2’s Comments

Regarding the article, I have next comments and recommendations:

  • Be careful about Latin names of fungal species throughout the article, they should be written in Italics, you have used it but throughout the article sometimes not, and this should be modified, like L. 52-53: Hymenochaetaceae family; L. 361: Hymenochaetaceae family; L. 642: the Fulvifomes sp. and P. boldo extracts; L. 676: the Inonotus clade. Likewise: L. 362-363, in capture of Figure 2: p-coumaric acid – p-should be in Italics; in Table 1 and capture of Figure 1: better: p-Coumaric acid; in capture of Figure 6 and 7: vitamin C and Fulvifomes sp.

We have corrected the format as requested.

  • L. 86:… derived from Peumus boldus have been attributed…

We have corrected in the manuscript, line 87.

  • Units: between value and °C should be a space, like 35 °C, 98 °C, 25 °C, 4 °C, etc. Correct throughout the article. L. 155: at 2500 rpm at 25 °C: convert rpm to g (rcf). L. 182: better: 0.8 M NaOH. You use for μg/ml, mg/mL and μL, etc., decide and unify only one way l or L throughout the article.

We have corrected the entire manuscript.

  • In used equipment complete where is needed: producer, town and country: NEX food dehydrator; electric blade grinder (Thomas, town, Germany); a centrifuge (Hettich Rotina 380, town, Germany); a microplate reader (VERSA Max, town or state, United States); a microplate reader (VERSA max, town or state, USA); a microplate reader (VERSA Max, town or state. USA); a fluorescence microscope (Nikon Ti-E, town, Japan).

We have added the city to each equipment.

  • In Equation 3: use English instead of Spanish: Abs sample and Abs blank.

We have corrected it to English.

  • In References you used a mixture of full journal titles and abbreviations. So, in References substitute journal full titles with their abbreviations where it is needed: Redox Biol.; For. Ecol. Manag.; Int. J. Biol. Macromol.; J. Tradit. Complement. Med.; J. Biosci. Bioeng.; J. Environ. Sci. Health; Mol. Biol. Evol.; Trends Biotechnol.; Anal. Biochem.; Microchem. J.; Food Res. Int.; Crit. Rev. Food Sci. Nutr.; Food Chem.; Waste Biomass Valori; Prep. Bioch. Biotechnol.; Oxidative Med. Cell. Longev.; Nat. Prod. Res.

We have corrected the abbreviations of the journals indicated in the bibliography.

Furthermore, an English-speaking research team member corrected English grammar and sentence structure throughout the manuscript.